# Characterization of Micelle Formation by the Single Amino Acid-Based Surfactants Undecanoic L-Isoleucine and Undecanoic L-Norleucine in the Presence of Diamine Counterions with Varying Chain Lengths

Amber Maynard-Benson [1], Mariya Alekisch [1], Alyssa Wall [2], Eugene J. Billiot [1], Fereshteh H. Billiot [1] and Kevin F. Morris [2],*

[1] Department of Physical and Environmental Sciences, Texas A&M University-Corpus Christi, 6300 Ocean Drive, Corpus Christi, TX 78412, USA; amaynard@islander.tamucc.edu (A.M.-B.); maleksich@islander.tamucc.edu (M.A.); eugene.billiot@tamucc.edu (E.J.B.); fereshteh.billiot@tamucc.edu (F.H.B.)

[2] Department of Chemistry, Carthage College, 2001 Alford Park Drive, Kenosha, WI 53140, USA; awall@carthage.edu

* Correspondence: kmorris@carthage.edu

**Abstract:** The binding of linear diamine counterions with different methylene chain lengths to the amino-acid-based surfactants undecanoic L-isoleucine (und-IL) and undecanoic L-norleucine (und-NL) was investigated with NMR spectroscopy. The counterions studied were 1,2-ethylenediamine, 1,3-diaminopropane, 1,4-diaminobutane, 1,5-diaminopentane, and 1,6-diaminohexane. These counterions were all linear diamines with varying spacer chain lengths between the two amine functional groups. The sodium counterion was studied as well. Results showed that when the length of the counterion methylene chain was increased, the surfactants' critical micelle concentrations (CMC) decreased. This decrease was attributed to diamines with longer methylene chains binding to multiple surfactant monomers below the CMC and thus acting as templating agents for the formation of micelles. The entropic hydrophobic effect and differences in diamine counterion charge also contributed to the size of the micelles and the surfactants' CMCs in the solution. NMR diffusion measurements showed that the micelles formed by both surfactants were largest when 1,4-diaminobutane counterions were present in the solution. This amine also had the largest mole fraction of micelle-bound counterions. Finally, the und-NL micelles were larger than the und-IL micelles when 1,4-diaminobutane counterions were bound to the micelle surface. A model was proposed in which this surfactant formed non-spherical aggregates with both the surfactant molecules' hydrocarbon chains and n-butyl amino acid side chains pointing toward the micelle core. The und-IL micelles, in contrast, were smaller and likely spherically shaped.

**Keywords:** NMR; diffusion; amino acid-based surfactant; counterion; critical micelle concentration

## 1. Introduction

Surfactants are amphiphilic molecules containing both a hydrophobic hydrocarbon tail and a polar or charged head group. When the concentration of surfactant in an aqueous solution exceeds the critical micelle concentration (CMC), the molecules aggregate into micelles with the hydrophobic tails pointing inward and the hydrophilic head groups facing the bulk aqueous phase. Surfactant solutions disperse oils and other nonpolar compounds in an aqueous solution since the latter solubilizes in the micelles' hydrophobic core. Surfactants are, therefore, used in the petroleum, pharmaceutical, detergent, and cosmetics industries [1].

There is an increased demand from manufacturers, consumers, industrialists, and environmentalists to develop green alternatives to surfactant molecules that can be exploited in

the above applications. Alternatives that are less toxic and that are produced from biomass or natural products instead of from petroleum are favored. Amino acid-based surfactants fit these criteria because they are biodegradable, antimicrobial, and can be prepared from renewable resources. They also can easily be synthesized, are environmentally friendly in nature, relatively inexpensive, and have various potential industrial applications, such as drug delivery. In an amino acid-based surfactant, a hydrocarbon tail is covalently bound to an amino acid or peptide headgroup [1–10].

Capillary electrophoresis (CE), NMR spectroscopy, and molecular dynamics simulations have been used to characterize amino acid-based surfactants [11–14]. For example, changes in the CMCs, micelle radii, and a fraction of micelle-bound counterion molecules have been studied as a function of solution pH. Molecular dynamics simulations have been used to identify the micelles' chiral binding sites and to investigate the intermolecular interactions between the surfactant headgroups and chiral analytes. An understanding of these properties is needed to make optimal use of these materials in commercial formulations and separation experiments with CE. In the latter, separations must be conducted above the micelles' CMC and micelle radii, charge and counterion binding govern the electrophoretic mobilities of the micelles through the capillary [15].

The surfactants examined here were undecanoic L-isoleucine (und-IL) and undecanoic L-norleucine (und-NL). Molecular structures are shown in Figure 1.

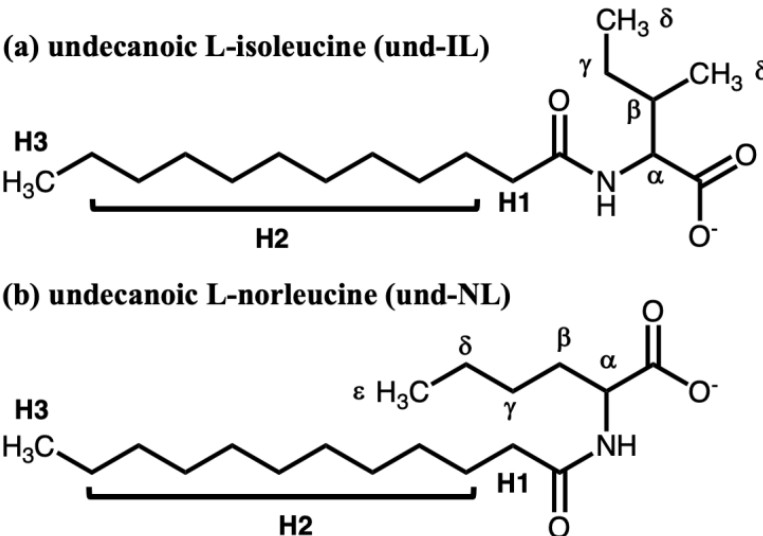

**Figure 1.** Chemical structures of (**a**) undecanoic L-isoleucine (und-IL) and (**b**) undecanoic L-norleucine (und-NL). Proton labels are used to reference the NMR resonances in Figure 2.

Isoleucine and norleucine are both isomers of leucine; however, only isoleucine is naturally occurring. Meats, fish, cheese, eggs, and seeds are good dietary sources of isoleucine [16]. Amino acid-based surfactants with an isoleucine headgroup have also been used as chiral selectors in the separation of racemic mixtures with CE [15]. Norleucine is a non-canonical amino acid used to study the structures and functions of proteins. It has also been detected in small concentrations in some bacteria [17,18]. The amino acid side chains in isoleucine and norleucine are sec-butyl and n-butyl functional groups, respectively. Both surfactants contain an eleven-atom hydrophobic tail and a hydrophilic amino acid head group. At the pH used in this study, the carboxylate is deprotonated, giving both surfactants a −1 charge.

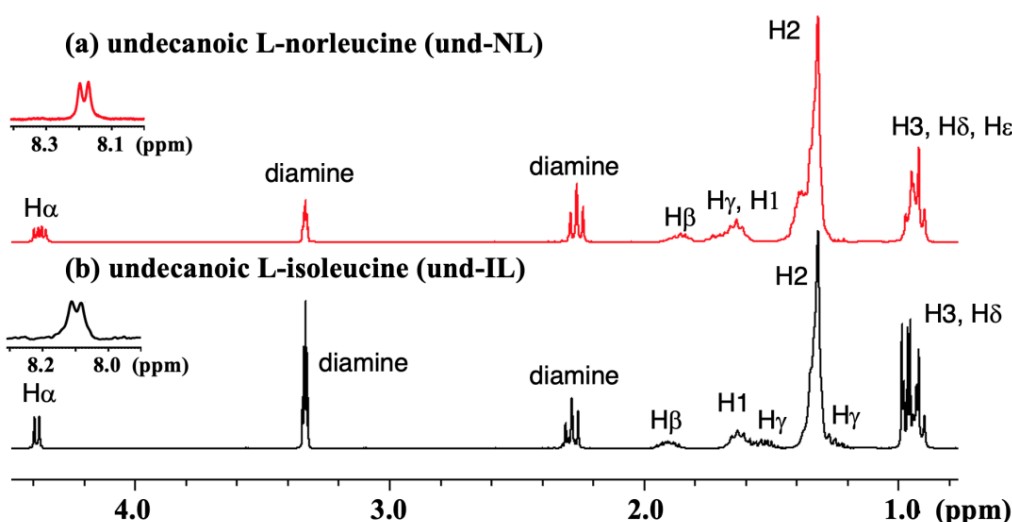

**Figure 2.** $^1$H NMR spectra of (**a**) undecanoic L-norleucine and (**b**) undecanoic L-isoleucine. Each surfactant's NH doublet is shown as an insert above the respective spectrum.

The goal of this study was to investigate how micelle counterions affected the physical properties of these two amino acid-based micellar systems. Counterions have previously been shown to affect the CMCs and radii of other anionic micelles. For example, Ali et al. studied sodium dodecyl sulfate micelles (SDS) in solutions containing L-Valine and L-Leucine. At low concentrations, the amino acid counterions decreased the surfactant's CMC, but at high concentrations, they bound strongly to the SDS monomers causing micelle formation to occur at a higher concentration [19]. Surface tension and conductivity measurements have also been used to measure SDS CMC values in solutions containing the amino acids L-Glutamic Acid, L-Tryptophan, and L-Histidine [20]. The effect of counterions on the micellization of amino acid-based surfactants has been investigated as well. Patra et al. studied the physiochemical properties of sodium N-dodecanoyl sarcosinate micelles in the presence of the counterions $Mg^{+2}$ and $SO_4^{-2}$. The surfactant's CMC was found to decrease with increasing salt concentration. In contrast, higher salt concentrations yielded a slight increase in the micelles' hydrodynamic radii [21]. Similarly, in experiments with the amino acid-based surfactant surfactin-$C_{16}$, which contains a hydrocarbon tail and a cyclic peptide headgroup, small, spherical micelles formed in solutions containing monovalent counterions, such as $Li^+$ and $K^+$. The divalent cations, however, $Ca^{+2}$ and $Mg^{+2}$ interacted strongly with the anionic surfactants and caused micelle radii to increase [22].

The counterions investigated here were linear diamines with two to six carbon methylene chains separating the ionizable amine functional groups. The structures of the diamines are shown in Table 1, along with each amine's pK$_a$ values. Sodium counterions were studied as well because $Na^+$ is often used as the counterion in separation experiments with amino acid-based micelles [15]. Including sodium counterions also allowed the behavior of the diamines, which had variable structures and charges, to be compared to a simpler +1 metal cation. Commercial formulations containing both surfactants and organic diamines have been reported. For example, Vu et al. showed that solutions containing the amino acid-based surfactant sodium lauryl sarcosinate were thickened by organic amine additives, such as L-Arginine, triethanolamine, and trimethylphenyl chloride. Thickening of these surfactant solutions is required for their application in personal care formulations [23]. In addition, linear diamines have been used as surfactant solution additives for applications in cosmetics [24], detergents [25], and oral pharmaceutical compositions for the delivery of drugs with low water solubilities [26].

**Table 1.** Structures and $pK_a$ values at 25 °C for the five diamine counterions investigated. $pK_a$s were taken from reference [27].

| Counterion | Structure | $pK_{a1}$ | $pK_{a2}$ |
|---|---|---|---|
| 1,2-ethylenediamine | $^+H_3N$ ⌃ $NH_3^+$ | 6.86 | 9.92 |
| 1,3-diaminopropane | $^+H_3N$ ⌄⌃ $NH_3^+$ | 8.88 | 10.55 |
| 1,4-diaminobutane | $^+H_3N$ ⌃⌄ $NH_3^+$ | 9.63 | 10.8 |
| 1,5-diaminopentane | $^+H_3N$ ⌃⌄⌃ $NH_3^+$ | 10.05 | 10.93 |
| 1,6-diaminohexane | $^+H_3N$ ⌃⌄⌃⌄ $NH_3^+$ | 10.76 | 11.86 |

The micelle physical properties examined here included the surfactants' CMC, the mole fraction of micelle-bound counterions, and the hydrodynamic radii of the micellar aggregates. The systems were characterized using NMR chemical shift analyses to determine the surfactants' CMC in mixtures containing different counterions. Diffusion Ordered Nuclear Magnetic Resonance Spectroscopy (DOSY) was used to measure the hydrodynamic radii of the micelles and to determine the mole fraction of micelle-bound counterions [28]. These experiments use pulsed magnetic field gradients to encode information about translational motion into NMR data sets. DOSY allows a diffusion coefficient, D, to be measured for each NMR-resolved component in a mixture. These data, in turn, can be used to resolve NMR spectra of the components in complex mixtures without carrying out physical separations [29–33].

## 2. The Materials and Methods

The diamines 1,2-ethylenediamine, 1,3-diaminopropane, 1,4-diaminobutane, 1,5-diaminopentane, and 1,6-diaminohexane, deuterium oxide (99.9% atom D), and tetramethylsilane were purchased from Sigma-Aldrich. Und-IL and Und-NL surfactants were synthesized by the method described by Lipidot et al. [34]. Surfactant purity was confirmed by comparing the relative integrals of the surfactant resonances in their NMR spectra. In addition, the NMR spectra of the recrystallized products showed no signs of impurities or starting material. Overall, surfactant purity was estimated to be greater than 98%. Annotated NMR spectra of both surfactants are shown in Figure 2. Solutions for NMR analysis were prepared gravimetrically. The pH of each solution was adjusted to 10.0 by adding small amounts of solid NaOH. All NMR measurements were made at this pH because surfactant solubility decreased considerably at more acidic pH values. Moreover, at pH 10, the counterions would be expected to be in cationic form.

All NMR spectra were collected on a Bruker 300 MHz spectrometer. The solvent was 90% $H_2O$/10% $D_2O$. The WATERGATE pulse sequence was used to suppress the solvent signal, and the number of scans were adjusted as needed based on the concentration of the analytes [35]. The temperature was maintained at 25 °C. Annotated NMR spectra of both surfactants are shown in Figure 2.

[1]H NMR was used to determine the CMC of surfactant solutions containing either und-IL or und-NL and each of the linear diamines listed above. In these experiments, NMR spectra were collected for a series of solutions whose concentrations were both above and below the surfactant's CMC. The chemical shift of the surfactant resonances did not change with increasing concentration below the surfactant's CMC. When the CMC was reached, however, surfactant molecules became incorporated into the micelles, and the electronic environment of the surfactant protons changed. As a result, the chemical shifts of the NMR resonances changed as well. The surfactant Hα and NH peaks behaved in

this manner. The concentration at which the chemical shift abruptly changed was the CMC [11]. Figure 3 shows a plot of the chemical shift of the und-IL NH resonance as a function of concentration. The counterion was 1,2-ethylenediamine. The CMC measured in this experiment was 12.2 mM.

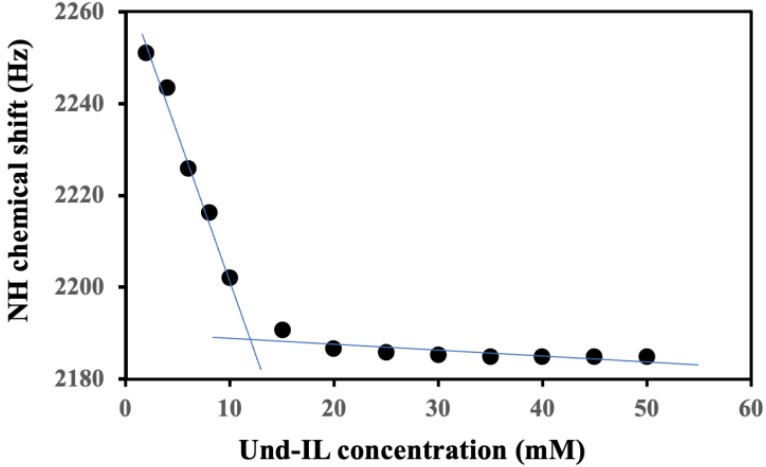

**Figure 3.** Plot of the chemical shift of the und-IL NH resonance as a function of surfactant concentration. The solution contained 1,2-ethylenediamine counterions.

Finally, it should be noted that all of the CMC measurements presented below were made using NMR spectroscopy. Other techniques used to determine surfactant CMC values include surface tension, conductivity, and fluorometry. Scholtz et al. compared CMC values for anionic, cationic, and neutral surfactants and found that the three above techniques gave very similar results [36]. Therefore, it is likely that the trends observed in the NMR-derived CMC values would also be seen if surface tension or conductivity measurements were used to determine the surfactants' CMCs.

The solutions prepared for the DOSY analyses contained 50.0 mM of either und-IL or und-NL and 50.0 mM of one of the counterions listed above. A total of 50.0 mM is well above each surfactant's CMC, so the surfactant molecules would be expected to primarily be in micellar form at this concentration [29]. Approximately 10 µL of tetramethylsilane (TMS) was also added to each NMR sample. Overall, the samples contained less than one percent by weight of the TMS probe. The solutions were mixed and allowed to equilibrate at 25.0 °C before NMR experiments were performed.

In the DOSY NMR experiments, separate resonances were resolved for the diamine counterion protons. The diffusion coefficient of the counterion could, therefore, be measured by monitoring the change in the intensity of these counterion peaks with increasing gradient strength. Tetramethylsilane molecules were used as probes to measure the diffusion coefficient of the micelles. TMS is very hydrophobic and partitions into and remains inside the micelle's hydrocarbon core. Therefore, in the DOSY experiment, the change in the intensity of TMS resonance with increasing gradient strength reports the diffusion coefficient of the entire micelle, $D_{micelle}$. [37–41]. In contrast, the surfactant molecules exchange between free solution and micelle-bound states. When micelle-bound, the surfactant molecules have a diffusion coefficient equal to $D_{micelle}$. In their free states, however, the diffusion coefficient of the surfactant molecules is much larger. Since exchange between these two states is fast on the NMR timescale, the surfactant diffusion coefficients measured with the DOSY experiment are the weighted average of their slower micelle and faster free solution D values. If micelle radii were calculated with these diffusion coefficients, the size of the micelles would be underestimated. Therefore, the TMS probe was used to directly determine $D_{micelle}$. An accurate measurement of the $D_{micelle}$ value is needed to calculate the micelles' hydrodynamic radii and the mole fraction of counterions bound to the micelles [10,13,29,38].

In each diffusion coefficient measurement, twenty NMR spectra were collected with increasing magnetic field gradient strength, G, using the bipolar pulse pair longitudinal encode–decode pulse sequence [42]. Typical G values ranged from 2.5 to 30.2 G·cm$^{-1}$. The intensity of the NMR resonances in these experiments decayed exponentially with a rate proportional to the quantity, $(\gamma \cdot G \cdot \delta)^2 \cdot (\Delta - \delta/3 - \tau/3)$, where $\gamma$ is the magnetogyric ratio, $\delta$ is the duration of the magnetic field gradient pulses, $\tau$ is the delay between the bipolar gradient pulses, and $\Delta$ is the diffusion time [42]. The $\Delta$, $\delta$, and $\tau$ values used in this study were 200, 4.0, and 0.20 ms, respectively. Linear plots of the natural log of the TMS and counterion peak intensities versus the quantity $(\gamma \cdot G \cdot \delta)^2 \cdot (\Delta - \delta/3 - \tau/3)$ were then prepared for each solution investigated. Linear regression analyses were used to calculate the slope of each plot. In each case, the slope equaled -D, where D is the diffusion coefficient. A representative plot for a solution containing und-IL and 1,4-diaminobutane is shown in Figure 4. A DOSY spectrum of a mixture containing 50 mM und-IL with Na$^+$ counterions is shown in the Supplemental Information.

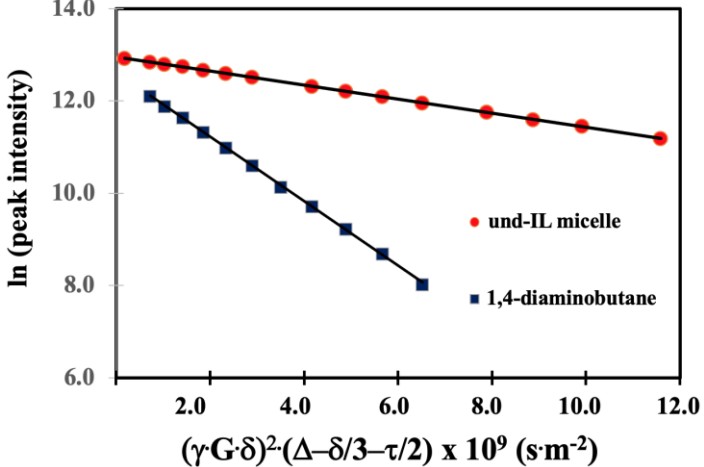

**Figure 4.** Plot of ln(peak intensity) versus $(\gamma \cdot G \cdot \delta)^2 \cdot (\Delta - \delta/3 - \tau/2)$ for a solution containing 50.0 mM und-IL and 50.0 mM 1,4-diaminobutane. The slope of each line is -D.

Analysis of the TMS intensity provided the diffusion coefficient of the micelle and D$_{micelle}$, and analysis of the counterion resonance intensity provided the observed diffusion of the counterion, D$_{obs, CI}$, in the micellar solutions. As discussed below, the latter is the weighted average of the counterion diffusion coefficient in its free and micelle-bound states. The chemical shifts of the counterion resonances relative to TMS used for these calculations were as follows: 1,2-ethylenediamine, 2.60 ppm; 1,3-diaminopropane, 2.90 ppm; 1,4-diaminobutane, 2.74 ppm; 1,5-diaminopentane, 2.73 ppm; 1,6-diaminohexane 2.76 ppm. Three trials of each diffusion experiment were performed.

Diffusion coefficients were then used to calculate the micelles' hydrodynamic radii, R$_h$, and the mole fraction of micelle-bound counterions f$_b$. Hydrodynamic radii were calculated using the Stokes–Einstein equation (Equation (1)), where D$_{micelle}$ is the micelle diffusion coefficient, $k_B$ is Boltzmann's constant, T is absolute temperature, and $\eta$ is the viscosity. A viscosity of 1.06 ± 0.02 cp reported by Lewis et al. was used in all radii calculations [13].

$$D_{micelle} = \frac{k_B \cdot T}{6 \cdot \pi \cdot \eta \cdot R_h} \tag{1}$$

The mole fraction of micelle-bound counterion, f$_b$, was calculated with Equation (2).

$$f_b = \frac{D_{obs(CI)} - D_{free(CI)}}{D_{micelle} - D_{free(CI)}} \tag{2}$$

Here, $D_{obs,CI}$ is the diffusion coefficient of the counterion in the presence of the micelles, $D_{free,CI}$ is the diffusion coefficient of the counterion in free solution, and $D_{micelle}$ is the micelle diffusion coefficient. Equation (2) holds because the counterion undergoes fast exchange on the NMR timescale between its free solution and micelle-bound states. When micelle-bound, the counterion diffuses at the same rate as the micelle [29,43]. Free solution counterion diffusion coefficients were measured by carrying out DOSY experiments with solutions containing 50.0 mM counterion and no surfactant.

### 3. Results and Discussion

Table 1 shows the structures and $pK_a$ values of the diamines used in this study [27]. The $pK_a$ values suggest that at pH 10, 1,2-ethylenediamine exists as a mixture of neutral and monoprotonated or +1 ions. The $pK_a$ for 1,3-diaminopropane is higher than 1,2-ethylenediamine, suggesting that at pH 10, this counterion is predominately monoprotonated or +1. Finally, at pH 10, the $pK_a$ values in Table I suggest that 1,4-diaminobutane, 1,5-diaminopentane, and 1,6-diaminohexane all exist as mixtures of +1 and +2 ions. In addition, moving from 1,2-ethylenediame to 1,6-diaminohexane, the diamine's linear methylene chain becomes longer, and the amines become more hydrophobic. The effect of counterion charge and hydrophobicity on the physical properties of the und-IL and und-NL micelles is examined below, beginning with a discussion of the CMC results and continuing with an examination of micelle radii, and the mole fraction of diamine counterions bound to the micelles. It should be noted that along with the NMR methods used here, other techniques, such as dynamic light scattering (DLS), are used to investigate the properties of micellar solutions [14,44]. For example, Arkhipov et al. used NMR diffusion experiments and DLS to measure the CMCs and solubilizing capacities of aggregates formed by rhamnolipid surfactants. Good qualitative agreement was seen between the NMR diffusion and DLS experiments. This study also highlighted that an important feature of the NMR diffusion experiments was that they allowed separate diffusion coefficients to be measured for each component in the mixture [44]. This feature is exploited here when studying the binding of the diamine cations in Table 1 to the und-IL and und-NL micelles.

### 3.1. Critical Micelle Concentrations

Table 2 shows CMC values for und-IL and und-NL micelles in solutions containing $Na^+$ and each of the diamine counterions. In the $Na^+$ counterion measurements, solutions contained surfactant and $NaHCO_3$. Uncertainties were calculated from three replicate measurements. Table 2 results show that the CMCs of both surfactants were higher in solution with $Na^+$ counterions and lower in solutions containing the diamines. For example, the und-IL and und-NL CMC values were 13.0 mM and 12.8 mM, respectively, when the counterion was $Na^+$. These values were higher than the CMCs measured for each solution containing the diamines. Table 2 also shows that the surfactant CMCs decreased when the length of the diamine methylene chain length increased. The und-IL CMC was 12.2 mM with 1,2-ethylenediamine counterions. As the length of the diamine methylene chain increased, the und-IL CMC decreased, reaching a minimum value of 2.0 mM with 1,6-diaminohexane. Similarly, the und-NL CMC was highest with 1,2-ethylenediamine (11.5 mM), decreased steadily with diamine methylene chain length, and reached a minimum value of 4.9 mM in a solution containing 1,6-diaminohexane. These results will now be compared to previous work with the amino acid-based surfactant undecanoic L-phenylanaline (und-Phe). In addition, the trends in Table 2 will be discussed in terms of the entropic hydrophobic effect, differences in counterion charge, and the ability of the linear diamine counterions to facilitate micelle formation by binding to multiple surfactant monomers.

A similar CMC trend was observed in an NMR study of the physical properties of undecanoic L-phenylanaline (und-Phe) amino acid-based surfactants in solutions containing $Na^+$, L-Arginine, and L-Lysine counterions [11]. L-Arginine contains a primary amine and a side chain guanidinium functional group, both of which are ionized in acidic, neutral, and slightly basic solutions. L-Lysine, such as the linear diamines studied here,

has two ionizable amine functional groups. The study of und-Phe micelles found that at pH 10, the surfactant CMC was 14.1 mM with $Na^+$ counterions but dropped to 5.7 mM and 6.2 mM in the presence of L-Arginine and L-Lysine, respectively. Therefore, like the linear diamines in Table 1, L-Arginine and L-Lysine facilitated micelle formation compared to $Na^+$. In this study, CMC values were measured for two amino acid-based surfactants with different side chains than und-Phe. Additionally, by measuring the surfactant CMCs with the counterions shown in Table 1, the spacer length between the amine functional groups could be systematically varied from two to six carbons, whereas, with L-Arginine and L-Lysine, the spacer length was fixed at four and five carbons, respectively.

**Table 2.** CMC values (mM) for Undecanoic L-Isoleucine and Undecanoic L-Norleucine at pH 10 and 25 °C.

| Counterion | Undecanoic L-Isoleucine CMC (mM) | Undecanoic L-Norleucine CMC (mM) |
|---|---|---|
| $Na^+$ | $13.0 \pm 0.1$ | $12.8 \pm 0.1$ |
| 1,2-ethylenediamine | $12.2 \pm 0.1$ | $11.5 \pm 0.8$ |
| 1,3-diaminopropane | $10.9 \pm 0.2$ | $9.0 \pm 0.1$ |
| 1,4-diaminobutane | $5.8 \pm 0.3$ | $7.2 \pm 0.5$ |
| 1,5-diaminopentane | $3.0 \pm 0.1$ | $6.1 \pm 0.7$ |
| 1,6-diaminohexane | $2.0 \pm 0.1$ | $4.9 \pm 0.9$ |

Both the basic amino acids L-Arginine and L-Lysine and linear diamines in Table 1 likely facilitate micelle formation by bridging between two surfactant monomers [11]. This behavior, shown in Figure 5 with 1,4-diaminobutane counterions, is not possible in $Na^+$-containing solutions where the counterion is a point +1 charge. The interactions shown in Figure 5, in turn, allow the counterions to act as templating agents, facilitating the aggregation of surfactant monomers and causing micelles to form at a lower concentration. Figure 5 behavior, however, would likely be less effective in 1,2-ethylenediamine and 1,3-diaminopropane where the amine functional groups are closer together and, thus, steric factors may prevent the amines from effectively binding to different surfactant monomers. This prediction is consistent with Table 2 results, which show that the surfactant CMCs are lower in solutions with diamines containing shorter methylene chains. For example, the und-IL CMC was 12.2 mM and 10.9 for 1,2-ethylenediamine and 1,3-diaminopropane, respectively. The und-NL CMC was 11.5 mM and 9.0 mM for 1,2-ethylenediamine and 1,3-diaminopropane, respectively. Diamine counterions with longer methylene chains, such as 1,4-diaminobutane, 1,5-diaminopentane, and 1,6-diaminohexane had lower CMCs likely because the steric factors discussed above are less important as the length of the methylene chain increases. Table 2 also shows that for both und-IL and und-NL the surfactant CMCs were lowest for the 1,6-diaminohexane counterion, which had the longest methylene chain.

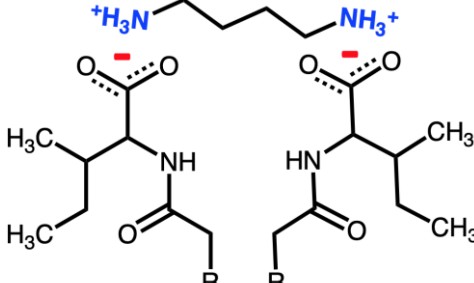

**Figure 5.** 1,4-diaminobutane templating or bridging between two und-IL surfactant monomers.

In addition, the $pK_a$ values in Table 1 indicate that a significant population of the 1,4-diaminobutane, 1,5-diaminopentane, and 1,6-diaminobutane ions in solution at pH 10 have a +2 charge. [43] Doubly protonated or +2 diamine counterions likely more readily form the structures shown in Figure 5 when compared to 1,2-ethylenediamine or 1,3-diaminopropane counterions which are predominately neutral or singly charged at pH 10.

Koyma also investigated the aggregation of octanoic, decanoic, lauric, and myristic fatty acids in the presence of $Na^+$, ammonium, and L-Arginine counterions. This study found that in fatty acid solutions containing bulkier counterions, such as L-Arginine, the CMC was lower than corresponding CMCs in solutions with smaller counterions, such as $Na^+$. The bulkier L-Arginine counterions were postulated to more effectively reduce the repulsion between anionic monomer headgroups at the micelle surface [43]. Table 2 results are consistent with the Koyma study in that the und-IL and und-NL surfactants have lower CMCs when in solution with bulkier diamine counterions compared to $Na^+$. In addition, as the size of the diamine counterion increased, the CMC decreased. This result is also consistent with the Koyma study [45].

Finally, when the methylene chain length of the counterion increases, the hydrophobicity of the counterion increases as well. This increase in overall diamine hydrophobicity likely contributes to the surfactants aggregating at lower concentrations. Chen et al. used variable temperature CMC measurements of homologous series of ionic and nonionic surfactants to calculate entropy and enthalpy changes associated with micelle formation [46]. The decrease in CMC observed as the length of the surfactant's hydrocarbon chain increased was attributed to the entropic hydrophobic effect, where micelle formation displaced water molecules surrounding the hydrocarbon chains into the more disordered bulk phase [46]. Table 2 shows that as the length of the diamine methylene chain increases, the surfactant CMC decreases. This observation may result from a similar entropic effect. The water molecules surrounding the diamine methylene chain are also likely displaced into the bulk phase when the counterion associates with the micelle. A longer diamine chain, such as 1,6-diaminohexane, causes a larger displacement of the ordered water molecules compared to the shorter chain counterions. Therefore, the resulting increase in entropy may make micelle formation more thermodynamically favorable for the solutions containing counterions with longer methylene chains [47,48].

### 3.2. Hydrodynamic Radii and Mole Fraction of Bound Counterions

While the CMC measurements report the concentration at which surfactants form micelles, they do not generally provide a detailed picture of the structure of the micellar aggregates above the CMC. Therefore, NMR spectroscopy was used to measure both micelle and counterion diffusion coefficients in solutions containing 50.0 mM of both surfactant and counterion. The diffusion coefficient, D, reports the rate of a particle's thermal motion in solution. D values can be used to calculate micelle radii and the mole fraction of counterion molecules bound to the micelles using Equations (1) and (2). These diffusion coefficient-derived values will be discussed in order to investigate the structures of the micelle-diamine complexes.

Table 3 reports the average hydrodynamic radii ($R_h$) of the und-IL and und-NL micelles in solutions containing $Na^+$ and each diamine counterion. $R_h$ is the radius of a hypothetical hard sphere that has the same diffusion coefficient as the particle studied [49]. Therefore, since $R_h$ is an apparent size of a dynamic, hydrated particle, the radius includes both solvent (hydro) and shape (dynamic) effects. Furthermore, $R_h$ values report radii for the entire micellar complex, which includes both the micelle-bound surfactant monomers and counterion molecules. Changes in $R_h$ with pH have been used previously to elucidate the mechanism of counterion binding to amino acid-based micelles [11,13,14].

Table 3 shows that the und-IL and und-NL hydrodynamic radii were 9.7 Å and 11.9 Å when $Na^+$ counterions were present at the micelle surface. These results are in good agreement with a study by Lewis et al. which investigated micelles formed by the amino acid-based surfactant L-undecyl-L-leucinate (und-Leu). Und-Leu has a C11 hydrocarbon chain, such as the surfactants investigated here, but the headgroup was an L-Leucine amino acid instead of isoleucine or norleucine [14]. The Lewis et al. study found that with sodium counterions, micelles formed by und-Leu had hydrodynamic radii of $9.3 \pm 0.1$ Å, in good agreement with the und-IL and und-NL radii in Table 3. Furthermore, dynamic light scattering experiments with 50 mM solutions of the und–Leu surfactant

yielded a similar $R_h$ value of $11.3 \pm 0.5$ Å. Finally, both NMR and dynamic light scattering experiments showed that with sodium counterions, the und–Leu micelle radius did not change appreciably with solution pH [14].

**Table 3.** Average hydrodynamic radii ($R_h$) and $f_b$ values for undecanoic L-Isoleucine and undecanoic L-Norleucine micelles at 25 °C and pH 10.

| Counterion | Und-IL $R_h$ (Å) | Und-IL $f_b$ | Und-NL $R_h$ (Å) | Und-NL $f_b$ |
|---|---|---|---|---|
| Na$^+$ | $9.7 \pm 0.2$ | n/a | $11.9 \pm 0.1$ | n/a |
| 1,2-ethylenediamine | $9.8 \pm 0.2$ | $0.22 \pm 0.01$ | $9.2 \pm 0.1$ | $0.24 \pm 0.01$ |
| 1,3-diaminopropane | $12.3 \pm 0.2$ | $0.34 \pm 0.01$ | $19.6 \pm 0.8$ | $0.10 \pm 0.01$ |
| 1,4-diaminobutane | $15.6 \pm 0.1$ | $0.39 \pm 0.01$ | $39.4 \pm 0.3$ | $0.45 \pm 0.02$ |
| 1,5-diaminopentane | $14.7 \pm 0.3$ | $0.29 \pm 0.01$ | $20.0 \pm 2.0$ | $0.28 \pm 0.02$ |
| 1,6-diaminohexane | $11.0 \pm 1.0$ | $0.34 \pm 0.02$ | $24.5 \pm 0.1$ | $0.32 \pm 0.01$ |

In addition, the und-IL and und-NL radii with sodium counterions in Table 3 are very similar to the radii of 9.8 Å and 9.2 Å for und-IL and und–NL micelles, respectively, in solutions containing 1,2-ethylenediamine counterions. For und–IL the $R_h$ value increased to 12.3 Å for 1,3-diaminopropane and 15.6 Å for 1,4-diaminobutane. The $R_h$ value then decreased slightly to 14.7 Å and 11.0 Å for 1,5-diaminopentane and 1,6-diaminohexane counterions, respectively. Therefore, the und–IL micelles are largest when 1,4-diaminobutane counterions were present in the solution and smallest in the presence of Na$^+$ counterions. Similar but more pronounced trends are observed for und–NL. The und–NL $R_h$ value was 19.6 Å with 1,3-diaminopropane counterions. $R_h$ increased to 39.4 Å with 1,4-diaminobutane counterions, and then decreased to 20.0 Å and 24.5 Å, respectively, in the presence of 1,5-diaminopentane and 1,6-diaminohexane. Therefore, as with und–IL, the und–NL micelles were largest when 1,4-diaminobutane was present in the solution.

Table 3 shows the $f_b$ values for the five diamine counterions binding to und–IL and und–NL micelles at pH 10. With 1,2-ethylenediamine, $f_b$ values were similar (0.22 for und–IL and 0.24 for und–NL micelles) for the two micelle systems. With 1,3-diaminopropane, $f_b$ increased to 0.34 for und–IL but decreased to 0.10 for und–NL micelles. $F_b$ values for diamines with four, five, and six methylene chains will be discussed in more detail below, but we note here that the mole fraction of bound counterions was generally higher for these counterions than for 1,2-ethylenediamine and 1,3-diaminopropane. The lower $f_b$ values observed for the diamines with shorter methylene chains are likely due in part due to the charge of the counterion. The pK$_a$ values in Table 1 suggest 1,2-ethylenediamine and 1,3-diaminopropane counterions are primarily +1 cations at pH 10, compared to the diamines with longer methylene chains which are a mixture of +1 and +2 cations. Therefore, the lower charge for 1,2-ethylenediamine and 1,3-diaminopropane likely causes these counterions to be less attracted to negative micelle surface, thus leading to smaller $f_b$ values.

For both micellar solutions, $f_b$ was largest with 1,4-diaminobutane counterions (0.39 for und–IL and 0.45 for und–NL). $F_b$ values then decreased for both micelles when 1,5-diaminopentane and 1,6-diaminohexane were present at the micelle surface. As discussed above, $R_h$ values were also highest with 1,4-diaminobutane counterions. Thus, the counterion with a methylene chain length of four carbons gives both the highest mole fraction of bound counterions and the largest hydrodynamic radius for both surfactants. A correlation between $R_h$ and $f_b$ is expected because as more surfactant monomers aggregate, the micelles become bigger, and $R_h$ increases. However, as more monomers pack into the micelle, the ionic headgroups repel, thus impeding further micelle growth. Counterions at the micelle surface balance this repulsion of the negative headgroups [37,40]. Thus, when $f_b$ is larger and more counterions are micelle bound, $R_h$ is generally larger as well. Furthermore, the mode of binding of the counterions to the micelles is likely important as well. This effect will now be explored.

With doubly charged counterions, such as 1,4-diaminobutane, 1,5-diaminopentane, and 1,6-diaminohexane, it is likely that the counterions bind parallel to the micelle surface as shown in Figure 6. In this manner, both charged amine functional groups interact with different surfactant monomers. Previous work has shown that L-Lysine binds und–Phe micelles in this manner [11]. 1,4 and 1,6-diaminohexane have been shown to bind parallel to the surface of undecyl-LL-leucine valanate micelles as well [13]. In addition, Figure 6 shows that the singly charged 1,2-ethylenediamine and 1,3-diaminopropane counterions may instead bind perpendicular to the micelle surface since only one of their amine functional groups is charged. Since the radii of both micelles are largest with 1,4-diaminobutane counterions, this counterion may have an optimal spacing between the amine functional groups that most effectively allows the counterion to bridge between two surfactant monomers, as shown in Figures 5 and 6. More effective bridging then provides greater coverage of the micelle surface with counterions and allows more surfactant monomers to pack into the micellar aggregates.

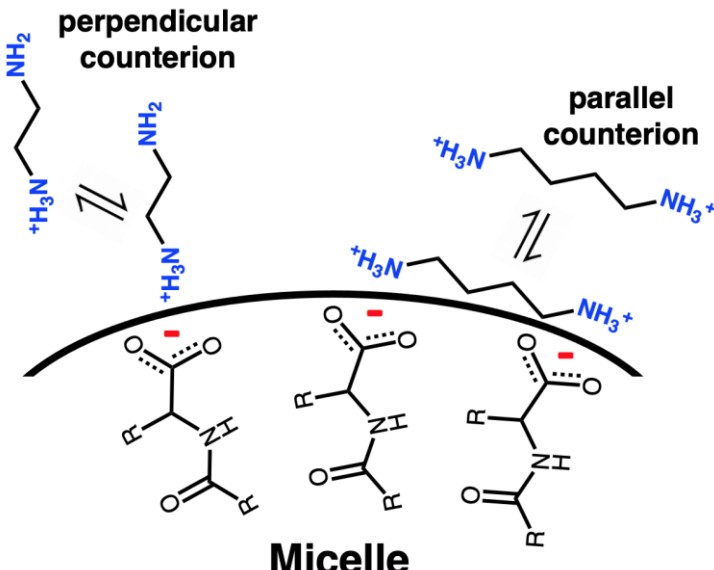

**Figure 6.** Model of und-IL or und-NL micelle surface showing perpendicular binding of 1,2-ethylene diamine counterions and perpendicular binding of 1,4-diaminobutane counterions.

All micelle radii and $f_b$ values reported in Table 3 were measured at pH 10. As discussed above, this pH was chosen because surfactant solubility decreased considerably at lower pH values. However, it would be expected that the diamine counterion would deprotonate above pH 10. As a result, the population of cationic counterions in the solution would decrease, and the counterions would be overall less attracted to the anionic micelle surface. To investigate this hypothesis, a DOSY experiment was performed with 50.0 mM und–IL and 50.0 mM 1,4-diaminobutane at pH 11.5. In this experiment, the micelle radius was $10.5 \pm 0.1$ Å. This value was very similar to the $11.0 \pm 0.1$ Å radius measured at pH 10. At higher pH, however, as expected, the mole fraction of micelle-bound counterions decreased from $0.39 \pm 0.01$ at pH 10 to only $0.10 \pm 0.01$ at pH 11.5.

Finally, Table 3 shows that at pH 10 with 1,4-diaminobutane counterions, the und–NL micelles are considerably larger than the und–IL micelles (15.6 Å for und-IL vs. 39.4 Å for und-NL). Each surfactant monomer has a hydrocarbon chain of the same length and a single terminal amino acid headgroup. The two surfactants, however, have different amino acid side chains. This difference likely contributes to the difference in $R_h$ values shown in Table 3. The larger radius observed for und–NL micelles may result from the surfactants' linear, n-butyl side chains turning toward the micelle core as shown in Figure 7.

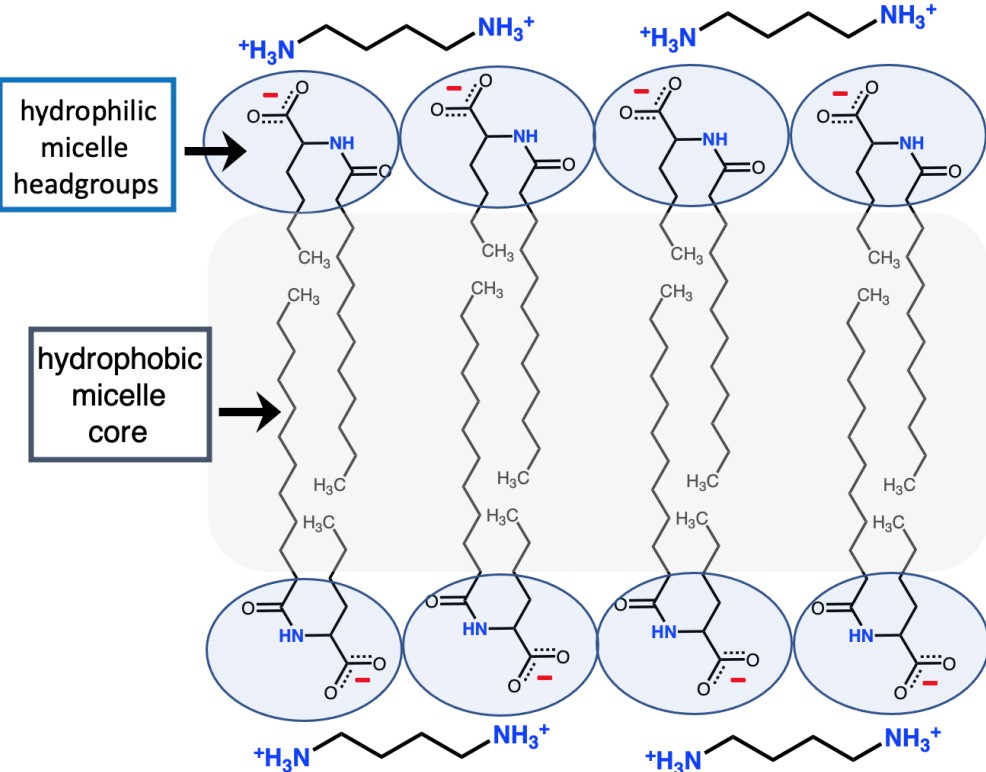

**Figure 7.** Model of und-NL micelles with 1,4-diaminobutane counterions.

The hydrophobic R-group of the und–NL monomers may be, therefore, behave somewhat like Gemini surfactants which have two hydrocarbon chains [50]. Perez et al. used X-ray scattering and NMR diffusion measurements to show that C11 Gemini surfactants with L-Arginine headgroups aggregated into cylindrical micelles. Good agreement between the X-ray scattering and NMR experiments was observed as well [51]. Similarly, the behavior shown in Figure 7 may cause the und–NL micelles to also form non-spherical aggregates. Finally, it should be noted that equation (2) reports the radius of the sphere with the same diffusion coefficient as the micelle. DOSY experiments alone, however, do not allow the exact shape of the micelles to be determined [49]. Further experiments, such as small angle X-ray scattering or dynamic light scattering, are needed to assess the exact shape of the micelles formed by und–NL with 1,4-diaminobutane counterions [52].

## 4. Conclusions

NMR measurements of und–IL and und–NL critical micelle concentrations in solution with linear diamine counterions showed that as the counterion methylene chain length increased, both surfactant's CMCs decreased. This observation was attributed to the counterions templating surfactant monomers below the CMC, the entropic hydrophobic effect, and differences in the charges of the counterions at pH 10. Furthermore, both und–IL and und–NL micelles had maximum hydrodynamic radii and a maximum mole fraction of micelle-bound counterions when 1,4-diaminobutane was present in the solution. It was hypothesized that this counterion's four-carbon methylene chain allowed it to most effectively bridge between neighboring surfactant monomers at the micelle surface. Finally, at pH 10, the und–NL micelles with 1,4-diaminobutane counterions were larger than the corresponding und–IL micelles with the same counterion. A model was proposed in which the und–NL monomers formed non-spherical micelles with both their hydrocarbon chains and amino acid side chains pointing toward the micelle core.

**Supplementary Materials:** The following supporting information can be downloaded at: https://www.mdpi.com/article/10.3390/colloids7020028/s1, Figure S1: DOSY spectrum of a mixture containing 50 mM und-IL at pH 10.

**Author Contributions:** Conceptualization, A.M.-B., E.J.B., F.H.B. and K.F.M.; methodology, A.M.-B., E.J.B., F.H.B. and K.F.M.; software, A.M.-B., M.A., E.J.B. and F.H.B.; validation, A.M.-B., M.A. and A.W.; formal analysis, A.M.-B., M.A. and A.W.; investigation, A.M.-B., M.A. and A.W.; resources, E.J.B., F.H.B. and K.F.M.; data curation, A.M.-B., M.A., E.J.B. and F.H.B.; writing—original draft preparation, A.M.-B., E.J.B. and F.H.B.; writing—review and editing, A.M.-B., E.J.B., F.H.B. and K.F.M.; visualization, A.M.-B., E.J.B., F.H.B. and K.F.M.; supervision, E.J.B. and F.H.B.; project administration, E.J.B., F.H.B. and K.F.M.; funding acquisition, E.J.B., F.H.B. and K.F.M. All authors have read and agreed to the published version of the manuscript.

**Funding:** This work was supported by National Science Foundation grants #1708959, #1709394, and #1709680, and the Robert A. Welch Chemistry Departmental Grant number BT-0041 awarded to the Texas A&M University Corpus Christi Chemistry Program.

**Data Availability Statement:** NMR spectra and supporting data are archived on a OneDrive at Texas A&M Corpus Christi. Contact Fereshteh Billiot (Fereshteh.Billiot@tamucc.edu) for access.

**Acknowledgments:** We acknowledge the generosity and support of the Ralph E. Klingenmeyer family.

**Conflicts of Interest:** The authors declare no conflict of interest.

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
