# Peer review of "Characterization of Micelle Formation by the Single Amino Acid-Based Surfactants Undecanoic L-Isoleucine and Undecanoic L-Norleucine in the Presence of Diamine Counterions with Varying Chain Lengths"

_colloids, doi:10.3390/colloids7020028_

Round 1

Reviewer 1 Report

Dear Authors, the current work is good but I had some technical queries and language/grammar concerns which can be found below:

 Abstract: The binding of linear diamine counterions with different methylene chain lengths to the amino-acid-based surfactants undecanoic L-isoleucine (und-IL) and undecanoic L-norleucine (undNL) was were investigated with NMR spectroscopy. The counterions studied were 1,2-ethylenediamine, 1,3- diaminopropane, 1,4-diaminobutane, 1,5-diaminopentane, and 1,6-diaminohexane. These counterions were all linear diamines with varying spacer chain lengths between the two amine functional groups. Sodium counterions were studied as well. All counterions investigated were linear diamines with difference in the spacer group, present in between the two functional groups. Suggestion: what functional groups, this should be clearly mentioned with the help of structure in graphical abstract.

 Sodium counterions were studied as well. Query: why sodium ions were also studied in this work or can say if sodium, why not other ions were taken into account. Kindly mentioned the requirement of use of Na??

“Results showed that the surfactants’ critical micelle concentrations (CMC) changed with the chain length of the diamine counterions, with the CMC decreasing as the counterion chain length increased”

Above sentence should be rewrite in a clearer way

“bringing together surfactant monomers”

What is the meaning of above statements? How the monomers brought together?? Not clear?

“The entropic hydrophobic effect and differences in diamine counterion charge were also 20 contributing factors”

Should rewrite like

The entropic hydrophobic effect and differences in diamine counterion charge were also contributed to the size of micelles and the value of CMC of the solution

“There is an increased demand from manufacturers, consumers, and Congress to develop green alternatives to the surfactant molecules used in the above applications”

Why congress? It seems inappropriate.

There is an increased demand from manufacturers, consumers, industrialists and environmentalists to develop green alternatives to the surfactant molecules that can be exploited in the above applications

“because many commercial surfactants pollute surface waters are derived from petroleum products.”

Above sentence should be rephrased

Amino acid-based surfactants fit these criteria because they are biodegradable, antimicrobial, and can be prepared from renewable resources. They are also can easily be synthesized, environmentally friendly in nature, relatively inexpensive and have various potential industrial applications like, …..,…..,….drug discovery etc. In an amino acid-based surfactant, a hydrocarbon tail is 46 covalently bound to an amino acid or peptide headgroup [1-8].

Capillary electrophoresis (CE), NMR spectroscopy, and molecular dynamic simulations have been used to characterize the physical properties of amino acid-based surfactants [9-16]. An understanding of these properties is needed to make optimal use of these materials in commercial formulations and separation experiments.

Understanding of which physical properties, Please state??

“To our knowledge, however, the effect of diamine counterions on the micellization of amino acid-52 based surfactants has not been systematically investigated.”

I we, our, like words should be avoided

“1H NMR was used to determine the CMC of surfactant solutions containing either und-IL or und-NL 101 and each of the linear diamines listed above.”

How can NMR be used to determine the CMC?

The chemical shift of a surfactant resonance 107 (typically H or NH) whose environment changes upon micelle formation was then 108 monitored. The concentration at which this chemical shift abruptly changed was the 109 CMC [27]. Figure 1 of the Supplemental Material shows a plot of the chemical shift of 110 the und-IL NH resonance as a function of concentration. The counterion was 1,2-111 ethylenediamine. The CMC measured in this experiment was 12.2 mM.”

Explain why the peak of NMR has been changed at CMC, peak area or shape can be change but peak position cannot be change…………Please explain

TMS is very hydrophobic and partitions into and remains inside the micelle’s 118 hydrocarbon core. In contrast, the surfactant resonances in 120 DOSY report the weighted average of the surfactant diffusion coefficient in its free 121 solution and micelle-bound states.

Not clear??

“Table 1 shows the structure and pKa values for the diamines used in this study. [34]”

Reference number should be used before the full stop

The Table 2 results show that the CMC’s of both surfactants were higher  in solution with Na+ counterions and lower in solutions containing the diamines. For  example, the und-IL and und-NL CMC values were 13.0 mM and 12.8 mM, respectively, 195 when the counterion was Na+. These values were higher than the CMC’s measured for  each solution containing the diamines.

Discussion for above results are required

“Rothbauer, et al. found that at pH 10, the 213 und-Phe CMC was 14.1 mM with Na+ counterions but dropped to 5.7 mM and 6.2 mM in 214 the presence of L-Arginine and L-Lysine, respectively. Therefore, like the linear 215 diamines, L-Arginine and L-Lysine facilitate micelle formation compared to Na+.  Rothbauer, et al. suggested that below the surfactant’s CMC, the amino acid counterions interacted with different surfactants molecules by bridging between two different 218 surfactant monomers [11].”

What was the requirement to add above mentioned reference. If this study was already done by Rothbauer what is the novelty in the  present research

Billot has selfcited his work 10 times which is ethically not good et al.

Reviewer 2 Report

In this work, the authors investigate the effect of diamine counterions on the aggregation of two bio-based anionic surfactants, namely und-IL and und-NL. The methodology is based on NMR for determining the CMC and DOSY NMR for the determination of hydrodynamic radii of aggregated objects. An interesting discussion is exposed regarding the influence of chain length in diamine counterions and their potential ability to bridge surfactant molecules together.

Some general remarks about the manuscript should be considered by the authors. Also, minor spelling mistakes were spotted in the manuscript.

Line 16: use the singular form of counterion.

Line 38: the chosen reference does not appear as the most relevant. The authors could cite a book like this one or a relevant review about surfactants in general.

Wieczorek, D.; Kwaśniewska, D. Novel Trends in Technology of Surfactants. In Chemical Technologies and Processes; Staszak, K., Wieszczycka, K., Tylkowski, B., Eds.; De Gruyter, 2020; pp 223–250.

Line 41: there might be a word missing “many commercial surfactants pollute surface water AND are derived from petroleum …” or “many commercial surfactants THAT pollute surface water are derived from petroleum …” but something is unclear about this sentence.

Line 62: I do not think that presenting the NMR spectra in the introduction is relevant. Moreover, the NH peak is not shown while the CMC determination is based on this peak shift only. It could be presented in the methods to illustrate how CMC is determined, along with the figure presented in supplementary materials.

Also, it would have been interesting to compare the surface tension measurement method for CMC determination with the NMR one. Moreover, this would provide information about the minimal surface tension, which is of interest for further applications involving these surfactants.

Line 96: How pure are the products? Has there been a quantification of impurities? It is well-known that even traces or impurities can have significant effects on aggregation behavior of surfactants, and for that reason it should be mentioned.

Paragraph 3.1: In this paragraph the plural form of CMC is indicated as “CMC’s” but should be “CMCs”. Check the entire section and the conclusion paragraph for corrections.

Line 261: The authors should change “hydrophobicity likely contributes the surfactants to aggregating at […]” into “hydrophobicity likely contributes to the surfactants aggregating at […]”.

Line 276 (section title): “Counterins” should be “counterions”.

Reviewer 3 Report

Dear Authors,

First, thank you for sharing your work: I really enjoyed reading it and I liked some of the ideas here proposed by you. However, I think that the article needs major revisions in order to increase its impact and can be improved after text revisions and new experiments when possible.

Here listed the points that should be reviewed in a new version of the manuscript:

11)      In the Abstract (Line 25) you claimed that your results suggested cylindrically shaped micelles just because you assessed longer diffusion time for such big micelles. You should better write that bigger micelles forms with und-NL and that you suggested a model that pass through the formation of non spherical micelles. There is no result that points clearly to this micellar shape.

22)      About the Introduction, I think some additional information should be added: previous work on counterions’ effects on CMC and on micelles’ shape and size. You also need to include something about the diamines, are they relevant as additives in formulation or other applications?   

33)      Materials and Methods (line 100): please add specific on the diffusion probe that has been used to perform DOSY experiments.

44)      Materials and Methods (line 116): please quantify the amount (w/w %) of TMS used in samples preparation.

55)      In all the manuscript you write “CMC’s” when you mean CMCs I assume (e.g., line 193).

66)      Results and discussion (line 347): to prove the model displayed in Figure 5 a titration on one of the 1,6-diaminohexane based systems will be great. When fully deprotonated, you should lose completely the bridging effect. I am still not sure why all the experiments have been conducted at pH=10 if there is an actual reason you should highlight that in the introduction.

77)      Conclusions (line 385): there are many ways to assess the cylindrical shape of micelles. One can perform SAXS or polarized DLS of course but also cheaper experiments can prove that. If you have cylinders and you increase the surfactant concentration (keeping the surfactant/counterion molar ratio constant) you should obtain a Hexagonal phase, while a Cubic phase is expected for spheres. The hexagonal phase will appear anisotropic under cross polarized light.

I hope that these suggestions would be useful for the authors in order to improve the quality of the article.

Reviewer 4 Report

The theme addressed in the manuscript is interesting but experimentally it is very poor. I believe that the interest of amino acid-derived surfactants is lacking to make the manuscript interesting for readers. In general, the manuscript has few experiments, they would have to perform a physicochemical characterization of the micellar interface to learn more about these micelles and show the interest of the systems under study.

On the other hand, they would have to do other experiments to arrive at the conclusions they show. In order to determine the size of the micelles and to be able to predict how the counterion is located at the interface, the authors would have to do DLS and zeta potential experiments. To determine the shape of the aggregates, the authors would have to do SAXS experiments.

In addition, the authors say that the variation of the CMC depends on the length of the hydrocarbon chain of the counterion, which is not something new. It could even happen that due to the length of the hydrocarbon chain, they change from direct micelles to vesicles. For example in the manuscript: Influence of the AOT counterion chemical structure in the generation of organized systems, LANGMUIR, 2020, 36, 10785 - 10793.

Regarding the bibliography, I think it is outdated, approximately 25% only corresponds to the last five years.

Round 2

Reviewer 1 Report

Dear authors

Happy to see that you have covered all the points in the revised version

Thanks 

Author Response

We are pleased to read this report and the Reviewer's positive recommendation.

Reviewer 3 Report

Dear Authors,

I really appreciated your punctual answers.

I think that the current article comes out improved and I am going to recommend it to be accepted as it is.

Best regards

Author Response

We are pleased to read this report and hear the Reviewer's positive recommendation.

Reviewer 4 Report

I continue to insist that experiments are lacking to achieve an adequate physicochemical characterization of micellar systems. DLS and zeta potential studies should be demonstrated to determine the presence of micelles and their surface charge. In this case, since the size is determined with the DOSY technique, it would serve to corroborate the results obtained, emphasizing the use of two techniques that give the same information but use different principles. I recommend doing DLS measurements, since the authors do not provide much information about the studied micellar systems, they only expose their formation. Authors would need to show DOSY spectra in the supplementary material.

The important thing to highlight is the use of the NMR technique to study these micellar systems. NMR is a very powerful technique as it provides a lot of information. I think the authors can get much more out of it. For example, perform a 13C and 14N spectroscopy and observe the shifts of the signals.
